



# Impact of International Shipping Emissions on Ozone and PM$_{2.5}$: The Important Role of HONO and ClNO$_2$

**Jianing Dai[1], Tao Wang[1]**

Department of Civil and Environmental Engineering, The Hong Kong Polytechnic University,

Hong Kong, 999077, China

*Correspondence to:* Tao Wang (cetwang@polyu.edu.hk)

**Abstract.** Ocean-going ships emit large amounts of air pollutants such as nitrogen oxide (NO$_x$) and particulate matter. Ship-released NO$_x$ can be converted to nitrous acid (HONO) and nitryl

chloride (ClNO$_2$), which produce hydroxyl (OH) and chlorine (Cl) radicals and recycle NO$_x$, thus affecting the oxidative capacity and production of secondary pollutants. However, these effects have not been quantified in previous investigations of the impacts of ship emissions. In this study, a regional transport model (WRF–Chem) revised to incorporate the latest HONO and ClNO$_2$ processes was used to investigate their effects on the concentrations of RO$_x$

(RO$_2$+HO$_2$+OH) radicals, O$_3$, and fine particulate matter (PM$_{2.5}$) in Asia during summer. The results show that the ship-derived HONO and ClNO$_2$ increased the concentration of RO$_x$ radicals by approximately two to three times in the marine boundary layer. The enhanced radicals then increased the O$_3$ and PM$_{2.5}$ concentrations in marine areas, with the ship contributions increasing from 9% to 21% and from 7% to 10%, respectively. The largest RO$_x$ enhancement was simulated

over the remote ocean with the ship contribution increasing from 29% to 50%, which led to increases in ship-contributed O$_3$ and PM$_{2.5}$ from 21% to 38% and from 13% to 19%, respectively. In coastal cities, the enhanced levels of radicals also increased the maximum O$_3$ and averaged PM$_{2.5}$ concentrations from 5% to 11% and from 4% to 8% to 4% to 12%, respectively. These findings indicate that modeling studies without considering HONO and ClNO$_2$ can

significantly underestimate the impact of ship emissions on radicals and secondary pollutants. It is therefore important that these nitrogen compounds be included in future models of the impact of ship emissions on air quality.



## 1 Introduction


Exhaust emissions by ocean-going ships affect the chemical compositions of the marine atmosphere and have a significant impact on the climate, air quality, and human health (Andersson et al., 2009; Corbett et al., 2007; Eyring et al., 2010; Liu et al., 2016). The key air pollutants emitted by ship vessels include gases such as sulfur dioxide ($SO_2$) and nitrogen oxides

($NO_x = NO + NO_2$) and particulate matter (PM) (Eyring et al., 2005; Moldanová et al., 2009). Emissions of $NO_x$ and other ozone ($O_3$) precursors (volatile organic compounds (VOCs) and carbon monoxide (CO)) from shipping contribute to the tropospheric $O_3$ burden and hydroxyl radicals (OH), thereby influencing global radiative forcing and oxidative power (Lawrence and Crutzen, 1999). Ship-generated aerosols also affect the radiative budget by scattering and

absorbing solar and thermal radiation directly and by altering cloud properties (Devasthale et al., 2006; Eyring et al., 2010; Fuglestvedt et al., 2009; Lawrence and Crutzen, 1999; Liu et al., 2016). Ship emissions in ports and near the coast also influence the air quality of coastal cities (Liu et al., 2018; Zhang et al., 2017c) and threaten public health (Campling et al., 2013; Liu et al., 2016). As international shipborne trade continues to increase, ship emissions are expected to

continuously grow at a rate of 3.5% over the 2019-2024 period (UNCTAD, 2019), and their impact on the environment is a growing concern.

The effects of ship emissions on the formation of $O_3$ and $PM_{2.5}$ have been extensively evaluated in numerical studies. On the open ocean, ship-generated $NO_x$ reacts with VOCs emitted from ships and from the background atmosphere and enhances $O_3$ formation (Aksoyoglu et al., 2016;

Corbett and Fischbeck, 1997; Huszar et al., 2010; Hoor et al., 2009; Lawrence and Crutzen, 1999). In coastal areas, the $O_3$ levels can also be increased by $NO_x$ emitted from ships in ports and harbors and through the dispersion of ship-formed $O_3$ on the open ocean (Aksoyoglu et al., 2016; Song et al., 2010; Wang et al., 2019). On the other hand, ship-generated $NO_x$ can reduce ozone formation via a titration effect in heavy-traffic ports and within the ship tracks (Aksoyoglu

et al., 2016; Wang et al., 2019). Ship emissions have also been shown to increase $PM_{2.5}$ concentrations via direct emissions and via the production of secondary aerosols through the reaction of gaseous precursors (Aksoyoglu et al., 2016; Liu et al., 2018; Lv et al., 2018). Although the formation of $O_3$ and secondary aerosols is affected by their precursors, it can also



be influenced by the levels of radicals, which are key to the oxidation of precursors. Limited
attention has been paid to the production of ship-related radicals in evaluating the effects of ship
emissions on secondary pollutants.

Recent studies have demonstrated the potentially important roles of two radical precursors and
nitrogen reservoirs—nitrous acid (HONO) and nitryl chloride ($ClNO_2$)—in the atmospheric
oxidation chemistry (Fu et al., 2019; Li et al., 2016; Sarwar et al., 2014; Simon et al., 2009;
Zhang et al., 2017b). HONO is emitted directly in combustion and soil (Kleffmann et al., 2005)
or produced by heterogeneous reactions of $NO_2$ on various surfaces (Finlayson-Pitts et al., 2003;
Monge et al., 2010; Ndour et al., 2008) and by photolysis of nitrate aerosol (Ye et al., 2016; Ye
et al., 2017). $ClNO_2$ is formed from reactions of $N_2O_5$, which is produced when $NO_2$ reacts with
$O_3$, on chloride-containing aerosol at night (Bertram and Thornton, 2009). Photolysis of HONO
and $ClNO_2$ by sunlight produces OH or Cl radicals and recycles $NO_2$, hence affecting the
oxidation capacity and production of secondary pollutants (Osthoff et al., 2008; Wang et al.,
2016). Ships can directly emit HONO (Sun et al., 2020), and their emitted $NO_x$ can produce
HONO and $ClNO_2$ via heterogeneous reactions on sea-salt and ship-emitted particles. Although
the production and effects of HONO and $ClNO_2$ from land-based emissions have been
demonstrated over land areas (Zhang et al., 2017a), few studies have examined the effects of the
two reactive nitrogen species from international shipping. Field studies have observed elevated
mixing ratios of HONO (0.2 ppb) at a marine site of the Bohai rim in northern China (Wen et al.,
2019) and of HONO (126 ppt) and $ClNO_2$ (1.97 ppb) at a coastal site in southern China (Tham et
al., 2014; Zha et al., 2014). These observations suggest the significant contribution of ship
emissions to the oxidative capacity of the marine and coastal atmosphere in East Asia.

In this study, we used a revised regional chemical transport model to simulate the spatial
distributions of HONO and $ClNO_2$ produced by ocean-going ships and their effects on the
formation of $O_3$ and $PM_{2.5}$ in East Asia, which has 8 of the world's top 10 container ports and the
world's most trafficked oceans. We selected the summer (July) of 2018 as the study period,
when the summer monsoonal wind prevails in Asia; together with the highest radiation and
temperature, it has the greatest impact on shipping during the year. We describe in Section 2 the
details of the model setting, emissions, numerical experiments, observational data, and model
validation. In Section 3, we exhibit the model performance for HONO and $ClNO_2$ and compare





the results with available measurements in marine areas; we then show the formation of ship-

related HONO and ClNO$_2$ and their subsequent effects on radicals, O$_3$, and PM$_{2.5}$ in oceanic

areas and coastal cities. Our conclusions are given in Section 4.

## 2. Methodology

### 2.1 Model Setting

In this study, the WRF–Chem model (version 3.6.1; Grell et al., 2005) with updated gases-phase

and heterogeneous-phase mechanisms of new reactive nitrogen species (Zhang et al., 2017a) was

used to simulate the transport, mixing, and chemical transformation of trace gases and aerosols.

Briefly, the updated module was based on the default CBMZ module (Zaveri and Peters, 1999),

in which the O$_3$ production came from the traditional photochemical mechanisms with only two

gas-phase sources of HONO (OH+NO $\rightarrow$ HONO and HO$_2$ + NO$_2$ $\rightarrow$ HONO) and no chlorine

chemistry. In the updated module (CBMZ-ReNOM), the HONO sources included the additional

gas-phase reactions between NO$_x$ and HO$_x$ (OH+HO$_2$), the heterogeneous reaction of NO$_2$ on the

particle, urban, leaf (Kurtenbach et al., 2001), and sea surfaces (Zha et al., 2014), and direct

emission from vehicles (Gutzwiller et al., 2002; Kurtenbach et al., 2001; Sun et al., 2020). For

this study, an additional HONO source from the photolysis of particulate nitrate was added to our

model. For ClNO$_2$ production, the parameterization from Bertram and Thornton (2009) was used

to represent the N$_2$O$_5$ uptake coefficient and ClNO$_2$ production yield. This parameterization

reproduced the order of magnitude and variation of the observed N$_2$O$_5$ and ClNO$_2$ levels in a

background site of Hong Kong (Dai et al., 2020). The other six chlorine species, with relevant

photolysis reactions and subsequent reactions between released Cl radical and VOCs, were also

added to the default module (Zhang et al., 2017a). The details of other chemical and physical

schemes for the simulation can be found in Zhang et al. (2017a).

The model simulations were performed from June 28 to July 31, 2018. The first 72 h of the

simulations were considered as a spin-up time. The initial meteorological conditions were

provided by reanalysis data from the final (FNL) Operational Global Analysis dataset provided

by the National Centers for Environmental Prediction (NCEP;



http://rda.ucar.edu/datasets/ds083.2/). The model had 31 vertical layers with a fixed top of 100 hPa. The domain covered a large part of Asia with a horizontal resolution of $36 \times 36$ km
(Figure 1a). The surface layer was 30 m above the ground, and the lowest 11 layers were approximately within the height of the planetary boundary layer at noon.

## 2.2 Emissions

Five sets of emission inventories (EIs) were used for anthropogenic emissions in our study. For
mainland China, the Multi-resolution Emission Inventory for China (MEIC; http://www.meicmodel.org/) in 2016 was used. For the rest of Asia, we applied the MIX (http://www.meicmodel.org/dataset–mix) for 2010 (Li et al., 2017). For international shipping, the emission database in the Community Emission Data System (CEDS) (McDuffie et al., 2020) for 2017 was used. The HONO emissions from land transportation sources were calculated using
land-based $NO_x$ emissions and the $HONO/NO_x$ ratios (0.8% for gasoline and 2.3% for diesel). For ship-emitted HONO, we set the emission ratio of $HONO/NO_x$ as 0.51% based on the reported ratio in fresh ship plumes in Chinese waters (Sun et al., 2020). For anthropogenic chloride emissions, the high-resolution ($0.1° \times 0.1°$) EIs of HCl and fine particulate $Cl^-$ for 2014 were applied for mainland China (Fu et al., 2018). These EIs included four sectors and have been
shown to offer a reasonable model simulation of particulate chloride by the WRF–Chem model (Dai et al., 2020). The Reactive Chlorine Emission Inventory (RCEI; Keene et al., 1999) was used for anthropogenic chloride emissions in the other regions. For natural emissions, the biogenic emissions were calculated by the Model of Emission of Gas and Aerosols from Nature (MEGAN) version 2.1 (Guenther et al., 2006).

The spatial distribution of ship $NO_x$ emissions is shown in Figure 1a. The main ship routes with high emission intensity are clearly identified in the ship emission inventory. One major shipping lane is located in the Southern Bay of Bengal (BOB) in the Indian Ocean, passes through the Strait of Malacca, and extends to the South China Sea (SCS) and other Asian countries. Distinct shipping lanes are also shown along the coast of the East China Sea (ECS) and the Sea of Japan
(SOJ). Over the West Pacific Ocean (WPO), congested ship routes are distributed among Japan and other countries (Southeast Asian countries, Australia, and North America). Based on the





distribution of ship NO$_x$ emissions, six water zones were selected, including three waters around China (SCS, ECS, Bohai Rim (BR)), two waters in other regions (SOJ and BOB), and one open ocean (WPO; Figure 1b). In addition, three densely populated city clusters were chosen (the

North Central Plain (NCP), the Yangtze River Delta (YRD), and the Pearl River Delta (PRD)).

## 2.3 Experimental Setting

Eight simulations were conducted with different emissions and chemistry, as listed in Table 1. In the Def and Def_noship cases, the WRF–Chem model was conducted with default chemistry

(i.e., the default CBMZ mechanism with only the two HONO sources and no chlorine chemistry). The differences between Def and Def_noship (i.e., Def-Def_noship) represent the effects of ship emissions with the default nitrogen chemistry. In the Cl and Cl_noship cases, an updated chlorine chemistry in the revised WRF–Chem model was used. The differences between the two cases (i.e., Cl-Cl_noship) represent the effects of ship emissions with the default and

additional chlorine chemistry. Similarly, in the HONO and HONO_noship cases, the additional HONO chemistry was used, and the difference between the two cases (i.e., HONO-HONO_noship) represents the default impact of ship emissions with additional HONO chemistry. In the BASE and BASE_noship cases, the integrated HONO and chlorine chemistry are considered. The differences between BASE and BASE_noship represent the shipping impact

with the integrated effects of HONO and chlorine species. The results from the BASE experiment will be used to validate the model performance.

## 2.4 Observational Data and Model Validation

Meteorological data from surface stations from NOAA's National Climatic Data Center (NCDC;

Figure S1a) comprising wind direction, wind speed, surface temperature, and specific humidity were used to validate model performance for the meteorological parameters. Conventional air pollutant data (NO$_2$, PM$_{2.5}$, and O$_3$) from surface stations (obtained from China's Ministry of Ecology and Environment; Figure S1b) were used to evaluate the simulated air pollutants over China's land areas. Table S2 summarizes the statistical performance of our model results. For

meteorological parameters, high R values (>0.85) and low mean bias (MB) indicate good





performance for the meteorological field. For regular air pollutants, the model overpredicted PM$_{2.5}$ (MB = 10.6 µg m$^{-3}$) and slightly underpredicted NO$_2$ (MB = 3.3 ppbv) and O$_3$ (MB = 5.5 ppbv). These biases in simulation can be partially explained by uncertainties in the model input, such as the land-use data (Dai et al., 2019) and emission inventory (Li et al., 2017).

The O$_3$ data from two remote sites (Ryori and Yonagunijima (Yona)) in Japan (https://www.data.jma.go.jp/ghg/kanshi/ghgp/o3) and one coastal background site in Hong Kong (Hok Tsui (HT)) were used to compare the model performance over the maritime areas. These three sites are located along the coasts of the SCS, ECS, and WPO regions (Figure 1b). As shown in Figure S2, with the default chemistry, the model underpredicted the O$_3$ mixing ratio at

the coastal and marine sites, with underestimation by 2.8, 4.8, and 2.3 ppbv at the HT, Ryori, and Yona sites, respectively (Table S3). With the addition of the HONO and ClNO$_2$ chemistry, the simulated O$_3$ levels at the marine sites were improved, with the MB from –2.8 to –1.5 ppbv at the HT site, –3.0 to –2.3 ppbv at the Ryori site, and –2.3 to –0.7 ppbv at the Yona site.

**3 Results**

**3.1 Simulated HONO and ClNO$_2$ and Contributions from Ship Emissions**

Figure 2a shows the horizontal distribution of the average HONO at the surface layer in the BASE case. The predicted HONO was widespread over the oceans, with mixing ratios ranging

from 0.005 to 0.300 ppbv and distinct higher concentrations along the main shipping lanes. The distribution of HONO was consistent with that of NO$_2$ (Figure S3) due to the homogeneous and heterogeneous conversion of NO$_2$ to form HONO. In the vertical direction, the simulated HONO was concentrated at the surface and reached up to 400 to 600 m in the coastal and marine areas (see Figure S4). Figure 3 shows the vertical profile of HONO from ship emissions in the nine

selected regions. Consistent with the overall HONO vertical distribution, ship-contributed HONO also peaked at the surface in the oceanic areas, with average HONO levels of 3 to 120 pptv in the MBL. The greatest contribution of ship emissions was simulated in the WPO (96%), followed by the SOJ (80%), the BOB (49%), three Chinese waters (14% to 16%), and the coastal





cities (3% to 12%). The varying ship contributions in these regions can be explained by the

relative strength of the emissions from ships and from the adjacent land areas.

High values of ClNO$_2$ were simulated along the coasts and peaked in the lower MBL (Figure 2b), with mixing ratios ranging from 2 to 400 pptv in oceanic areas and the highest value in the BR region. This distribution was in line with that of its precursors N$_2$O$_5$ (Figure S5) and particulate chloride (Figure S6). Vertically, the peak value of ClNO$_2$ was simulated in the

residual layer (100 to 300 m; Figure 3), with mixing ratios of 8 to 350 pptv in oceanic areas. Similar to HONO, the greatest ship contribution to ClNO$_2$ was also simulated in the WPO (61%), followed by other oceanic areas (9% to 24%) and coastal cities (5% to 11%).

We compared the modeled HONO and ClNO$_2$ with field observations made at some coastal and marine sites (see Table S4). The simulated HONO mixing ratios were 0.1 to 0.3 ppbv and 0.01 to

0.1 ppbv over BR and SCS, respectively, which were comparable with the measurements at the marine sites of BR (0.2 ppbv) (Wen et al., 2019) and SCS (89 pptv) (Table S4). For the coastal areas of other Asian countries, the simulated HONO compared well with the measurements in South Korea (0.60 ppbv) (Kim et al., 2015) and Japan (0.63 ppbv) (Takeuchi et al., 2013). HONO was simulated at approximately 5 pptv in the open ocean and 10 to 25 pptv along the

main shipping lanes (Figure 2a), which were comparable to the measured HONO (3 to 35 pptv) in the open ocean in Europe and North America (Kasibhatla et al., 2018; Meusel et al., 2016; Ye et al., 2016). For ClNO$_2$, the order of magnitude and variation of the measured N$_2$O$_5$ and ClNO$_2$ levels at the HT site have been reasonably reproduced by our model for early autumn of 2018 (Dai et al., 2020). The model performance of HONO and ClNO$_2$ in the land areas of mainland

China for the summer of 2014 was also evaluated by Zhang et al. (2017). Overall, our model ability in simulation of HONO and ClNO$_2$ is acceptable, and the model results are sufficiently reliable for further analysis.

### 3.3 Impact of Ship-Derived HONO and Chlorine on RO$_x$, O$_3$, and PM$_{2.5}$



In this section, we evaluate the ship effects on the main atmospheric radicals ($RO_x$,
     $OH+HO_2+RO_2$), $O_3$, and $PM_{2.5}$ with the default chemistry (described in Section 2.3) and with the
     additional HONO and chlorine chemistry.

### 3.3.1 $RO_x$

Figure 4 shows the simulated differences in the average daytime $RO_x$ mixing ratios at the surface
     from the cases with and without ship emissions using different chemistry. The $RO_x$ mixing ratio
     was noticeably increased by ship emissions over oceanic areas, and this enhancement was
     magnified by the additional nitrogen chemistry. With the default chemistry (Figure 4a), the
     average ship contribution to $RO_x$ was about 18% over the whole oceanic area. The addition of
the HONO and chlorine chemistry increased the ship contributions to 28% (Figure 4b) and 22%
     (Figure 4c), respectively. Photolysis of ship-generated HONO and $ClNO_2$ released radicals (OH
     and Cl; Figure S7) and recycled $NO_x$, which then oxidized VOCs and gave rise to high levels of
     $RO_x$. With the combined HONO and chlorine chemistry, the ship contribution was further
     increased to 38% (Figure 4d). This combined ship contribution was smaller than the sum of that
from the separate HONO and chlorine chemistry (22% + 28% = 50%), which can be explained
     by the nonlinear interactions of the chemical system. Figure 5 shows the vertical profile of the
     $RO_x$ mixing ratio from ship emissions in the nine regions. The enhanced $RO_x$ reached an altitude
     of greater than 2 km over the oceanic regions, indicating the significant impact of ship-derived
     HONO and $ClNO_2$ on the oxidative capacity in the marine troposphere.

The largest increase in the ship contribution to $RO_x$ was predicted in the WPO region (from 29%
     to 50%; Figure 9a), followed by other oceanic areas (from 3% to 12% to 6% to 17%) and coastal
     cities (from –2% to 3% to 4% to 6%). The maximum ship contribution in the WPO region was
     consistent with the greatest ship contribution to HONO and $ClNO_2$ in this region (Figure 4d). In
     the SCS and BOB regions, the enhanced $RO_x$ was more dispersed with the combined nitrogen
chemistry than that with the default and separate nitrogen chemistry. In the coastal cities, the
     $RO_x$ mixing ratio was also affected by ship emissions via the transport of ship-generated HONO
     and $ClNO_2$ by summertime winds.



### 3.3.2 O$_3$

Figure 6 shows the simulated differences in the average O$_3$ at the surface from the cases with and without ship emissions. Consistent with the impact of ship emissions on oceanic RO$_x$, the oceanic O$_3$ was also noticeably increased (by 9%) by ship emissions, which was further enhanced by the addition of HONO (12%) and ClNO$_2$ (14%) and combined nitrogen chemistry (21%). The simulated distribution of ship-enhanced O$_3$ with the default chemistry was along the

main shipping routes with high NO$_x$ emissions (Figure 6a). O$_3$ formation was highly sensitive to NO$_x$ from ship emissions due to the relatively low concentrations of NO$_x$ in the marine areas. The larger ship contribution with ClNO$_2$ chemistry than with HONO chemistry may be partially explained by a higher production ozone efficiency by NO$_2$ than by NO (from photolysis of HONO) and by the faster reaction rate of Cl radicals than OH radicals with long-lived alkanes.

With the combined impact from HONO and ClNO$_2$, widespread ozone increases were simulated over the SCS, BOB, and WPO regions (Figure 6d). The combined nitrogen chemistry also increased the ozone concentrations in the coastal areas; in contrast, these concentrations were decreased by HONO or ClNO$_2$ separately. As shown in Figure 6b and c, distinct ozone enhancement was simulated over the marine area of South Korea and Japan by HONO or ClNO$_2$.

However, this enhancement was weakened and even canceled by their combined effects. We calculated an indicator of the ozone formation regimes based on the ratio of the production rate of H$_2$O$_2$ to that of HNO$_3$ (P$_{H2O2}$/P$_{HNO3}$) (Fu et al., 2020; Zhang et al., 2009). Figure 7, with the combined HONO and ClNO$_2$ chemistry, shows that the NO$_x$-sensitive regime in east Asia was changed to a VOC-sensitive regime, which was probably due to the increase level of NO or NO$_2$

from photolysis of HONO.

Figure S8 shows the vertical profile of ship-generated O$_3$ enhancement in the nine regions. Similar to ship-enhanced RO$_x$, ship-related O$_3$ enhancement stretched from the surface to the lower troposphere (>2 km) over the marine regions. Because the emissions from ships occur at the sea surface, the vertically enhanced O$_3$ formation was caused by strong convection (Dalsøren

et al., 2009)


The ship contribution to O$_3$ formation was also simulated in the WPO region (from 21% to 38%; Figure 9b). In other oceanic areas, the contributions of ship emissions were also increased from 3% to 18% to 12% to 24%, with two distinct O$_3$ enhancements over the BR (~10 ppbv; Figure 6d) and ECS (15 ppbv) regions. In the three coastal city clusters, the reduced O$_3$ formation was

simulated by ship emissions with the default chemistry (from –5 to –1 ppb). Because these coastal cities are in the VOCs-limited regime (Figure 7a), the NO$_x$ from ships would lead to a decrease in chemical O$_3$ production. With the combined HONO and ClNO$_2$ effects, the ship-induced O$_3$ increased to –1 to –5 ppb due to the enhanced radicals and the transport of O$_3$ in the marine areas by ship-generated HONO and ClNO$_2$. The maximum O$_3$ increase in coastal cities

also doubled from 3 ppb (5%) to 7 ppb (11%), aggravating the negative effects of ship emissions on human health in coastal cities.

### 3.3.3 PM$_{2.5}$

Ship-derived HONO and ClNO$_2$ also influence the production of aerosols via changes in radicals

and NO$_x$. Figure 8 shows the simulated differences in the average PM$_{2.5}$ at the surface for cases with and without ship emissions. The PM$_{2.5}$ concentration was considerably enhanced by ship emissions, and the additional nitrogen chemistry further increased the simulated PM$_{2.5}$ concentration. With the default chemistry, the average ship contribution to the PM$_{2.5}$ concentration was about 7% in oceanic areas (Figure 8a), and it was increased to 10% with the

addition of HONO and ClNO$_2$ chemistry (Figure 8d). The greatest contribution from shipping to the PM$_{2.5}$ concentration was also simulated over the WPO region, as with ozone, with the contribution ranging from 13% with the default chemistry to 19% with the improved chemistry (Figure 9c). In other oceanic areas, the ship contributions were also increased from 2% to 12% to 6% to 15%. We calculated the effects of ship-generated HONO and ClNO$_2$ on the formation of

secondary particles. The additional ship HONO and chlorine chemistry increased the ship contribution to particulate nitrate from 13% to 41% in oceanic areas (Figure S9 and Figure S11) and its contribution to particulate sulfate from 11% to 34% (Figure S10 and Figure S11). In the coastal cities, the default contribution by ships was about 4% to 8%, which was also increased to 4% to 12% with the improved chemistry (Figure 9c). The considerable increase in ship-





contributed $PM_{2.5}$ and ozone due to HONO and $ClNO_2$ demonstrates the need to consider these compounds in evaluations of the impact of shipping on air quality.

Previous studies have evaluated the impact of ship emissions on the formation of $O_3$ and $PM_{2.5}$. Aksoyoglu et al. (2016) simulated the average ship contribution to oceanic $O_3$ as 10% to 20% in the Mediterranean area, and Huszar et al. (2010) showed a ship contribution of 10% over the

Eastern Atlantic. The maximum $O_3$ enhancement by ship emissions was 15 ppb in the coastal waters of South Korea (Song et al., 2010) and 30 to 50 $\mu g\ m^{-3}$ off the coast of the YRD region (Wang et al., 2019). For $PM_{2.5}$, the average ship contributions were about 20% to 25% in European waters (Aksoyoglu et al., 2016) and 2.2% to 18.8% off the coast of China (Lv et al., 2018).

Compared to previous studies, our study simulated a higher contribution to average ozone formation and a smaller contribution to average $PM_{2.5}$. However, it is difficult to discern the differences in these modeling studies due to the differences in the methodologies adopted, including ship emission inventory, model resolution, chemical mechanisms (in addition to different treatment of HONO and $ClNO_2$ chemistry), and period of study. All studies

demonstrate an important impact of ship emissions on atmospheric chemistry and air quality. The key finding of our study is the role of HONO and $ClNO_2$ in driving the oxidation processes, which has not been fully considered in most previous model studies of the impact of shipping on pollutant levels.

**4 Conclusions**

This study evaluated the production of HONO and $ClNO_2$ from international shipping and their impact on the oxidative capacity, ozone level, and level of fine PM in the maritime and coastal areas of eastern Asia. The results show that photolysis of the two compounds releases OH and Cl radicals, recycles $NO_x$, and increases conventional hydroxyl and organic peroxy radicals ($RO_x$ =

OH + $HO_2$ + $RO_2$) by 0.8% to 21.4%, $O_3$ by 5.9% to 16.6%, and $PM_{2.5}$ by 3.2% to 8.6% at the surface in coastal and Western Pacific regions. Their impact extends to the marine boundary layer. The largest contributions of HONO and $ClNO_2$ occur in the relatively remote oceans. Because ocean-going ships are a major source of $NO_x$, which is the key chemical precursor to





HONO and ClNO$_2$, it is important to consider the sources and chemistry of these nitrogen
compounds in evaluations of the impact of ship emissions.

**Code and data availability.** The codes and data used in this study are available upon request
from Tao Wang (cetwang@polyu.edu.hk).

**Author contributions.** TW initiated the research, and JD and TW designed the paper
framework. JD ran the model, processed the data, and made the plots. JD and TW analyzed the
results and wrote the paper.

**Acknowledgments.** We would like to thank Qiang Zhang from Tsinghua University for
providing the emission inventory and Xiao Fu from The Hong Kong Polytechnic University for
providing the code of HONO sources and anthropogenic chloride emission inventory.

**Financial support.** This research has been supported by the Hong Kong Research Grants
Council (grant no. T24-504/17-N) and the National Natural Science Foundation of China (grant
no. 91844301).

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

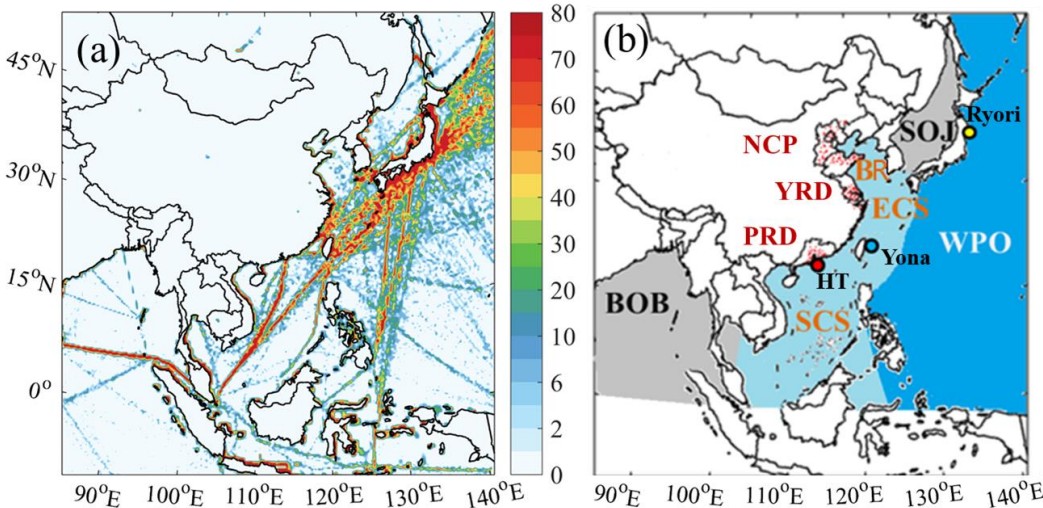

Figure 1: (a) NO$_x$ emission fluxes from ships (Unit: g m$^{-2}$ month$^{-1}$) in July 2017. (b) Model domains with six water zones (South China Sea (SCS), East China Sea (ECS), Bohai rim (BR), Sea of Japan (SOJ), Bay of Bengal (BOB), and West Pacific Ocean (WPO)), three coastal city clusters (Pearl River Delta (PRD), Yangtze River Delta (YRD) and North Central Plain (NCP)), and three maritime observational sites (Hok Tsui (HT), Yonagunijima (Yona), and Ryori). Red dots in PRD, YRD, and NCP represent selected coastal sites. Ranges of latitude and longitude in each body of water are listed in Table S1 in the supplementary information.





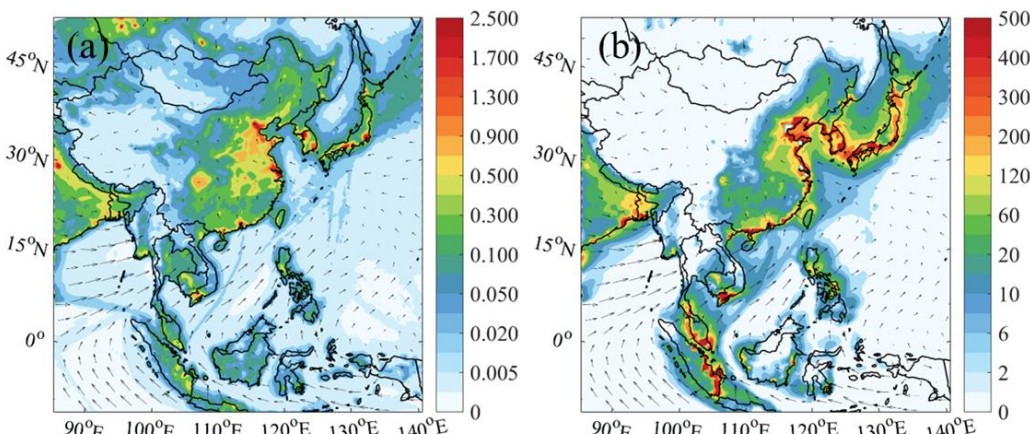

**Figure 2:** **Spatial distributions of the simulation of average (a) HONO (Unit: ppbv) and (b) nighttime ClNO₂ (Unit: pptv, 18:00 – 06:00 Local Standard Time (LST)) at the surface layer (~30 m) in July 2018 from the BASE case. Arrows present simulated wind vectors from the BASE case.**



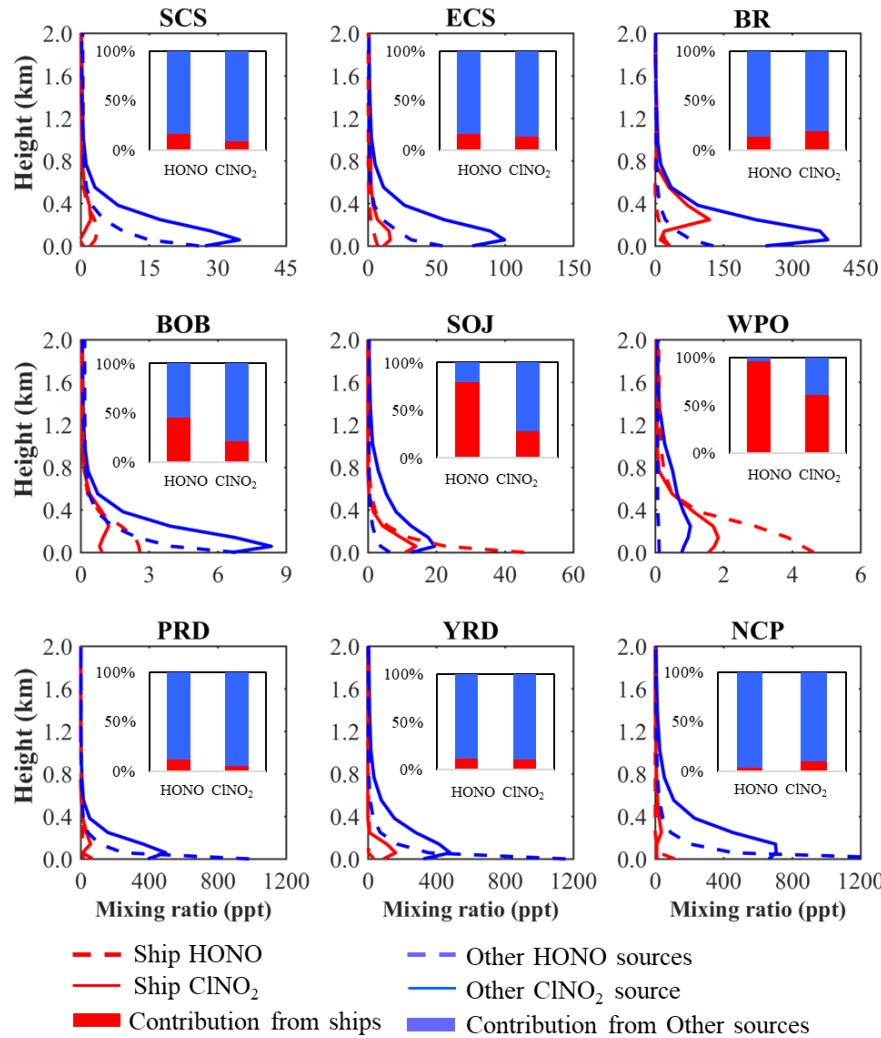

**Figure 3: Vertical profiles of simulated HONO and nighttime ClNO₂ (Unit: pptv) from ship emissions and other sources in nine regions. Also shown are contributions of ship emissions and other sources to average HONO and nighttime ClNO₂ levels in the marine boundary layer (within 600 m).**

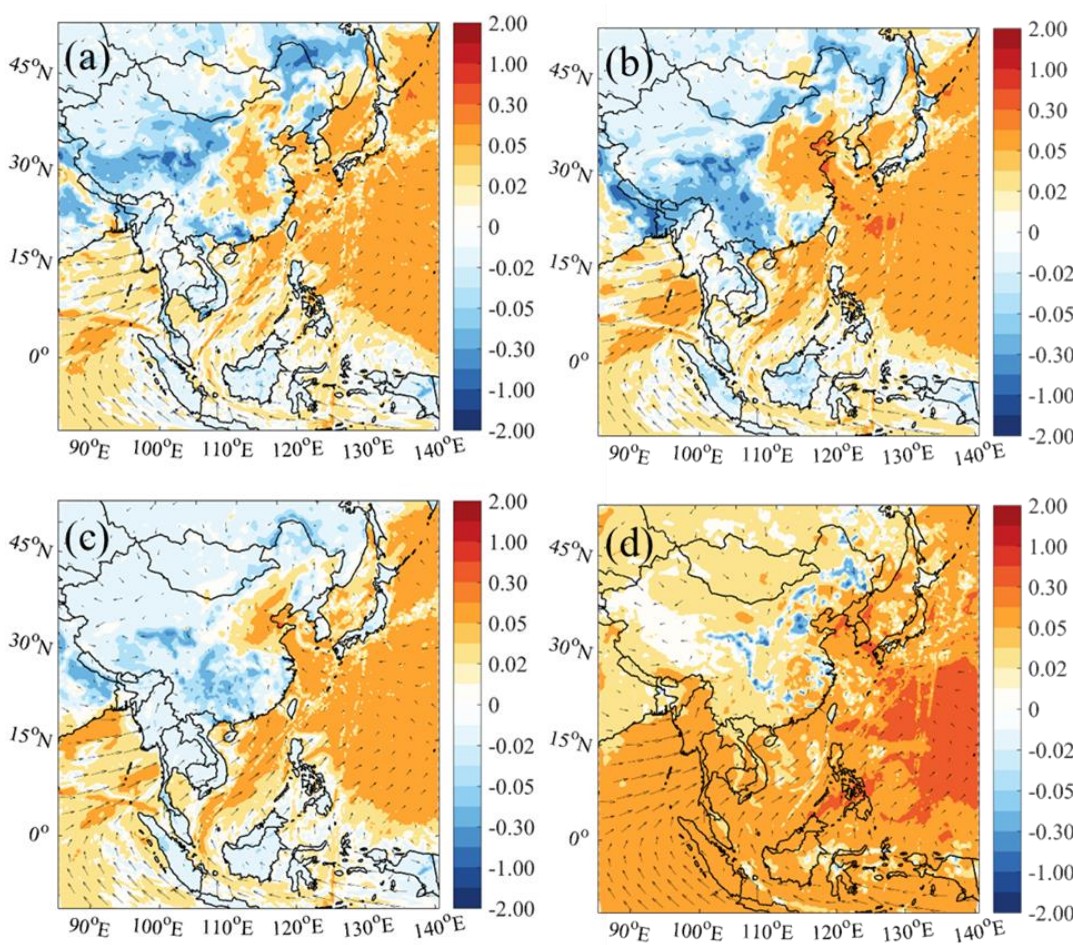

**Figure 4: Average daytime RO$_x$ variations (06:00-18:00 LST; Unit: pptv) with (a) default chemistry (Def-Def_noship), (b) default and additional HONO chemistry (HONO-HONO_noship), (c) default and additional chlorine chemistry (Cl-Cl_noship), and (d) default and combined HONO and chlorine chemistry (BASE-BASE_noship). Arrows present simulated wind vectors from BASE case.**





**Figure 5: Vertical profiles of daytime RO$_x$ variations (Unit: pptv) from different chemistry in nine regions.**



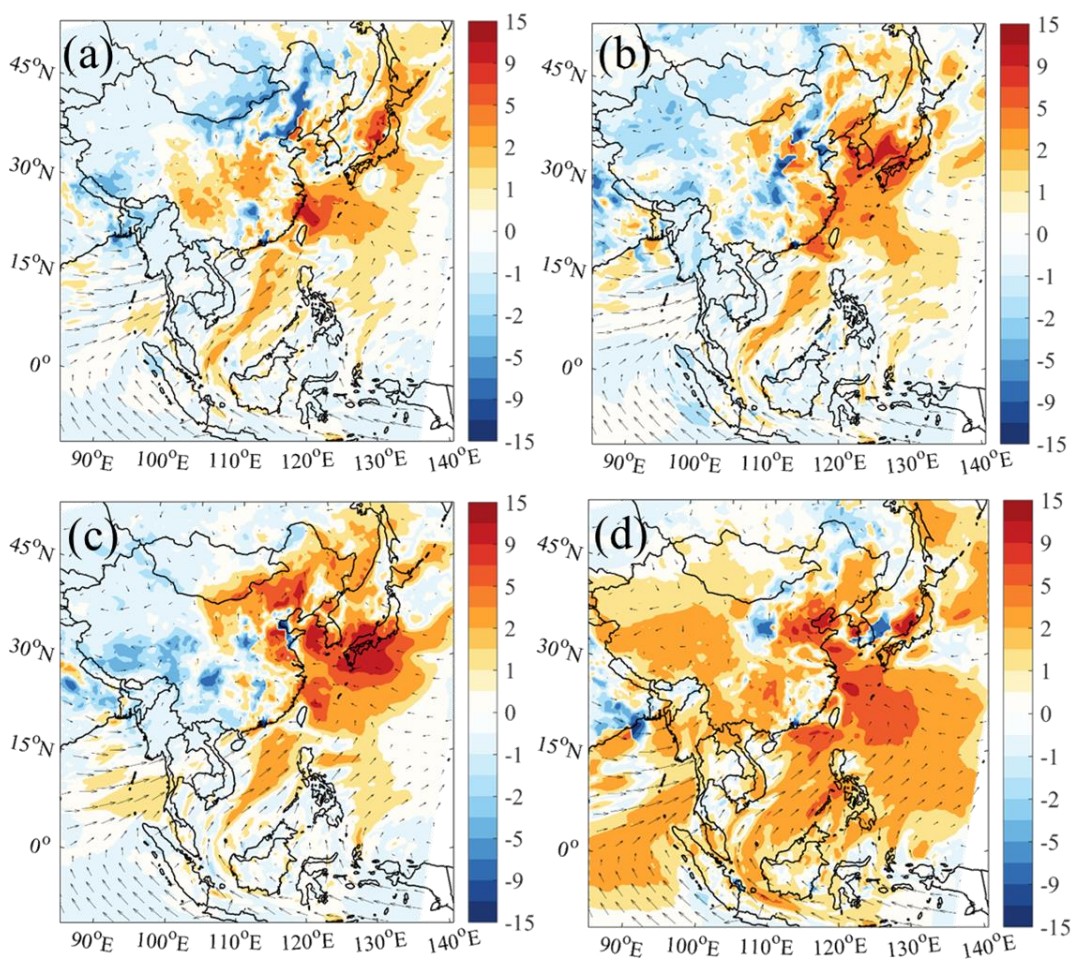


**Figure 6: This figure is the same as Figure 4 but for average ozone variations (Unit: ppbv).**




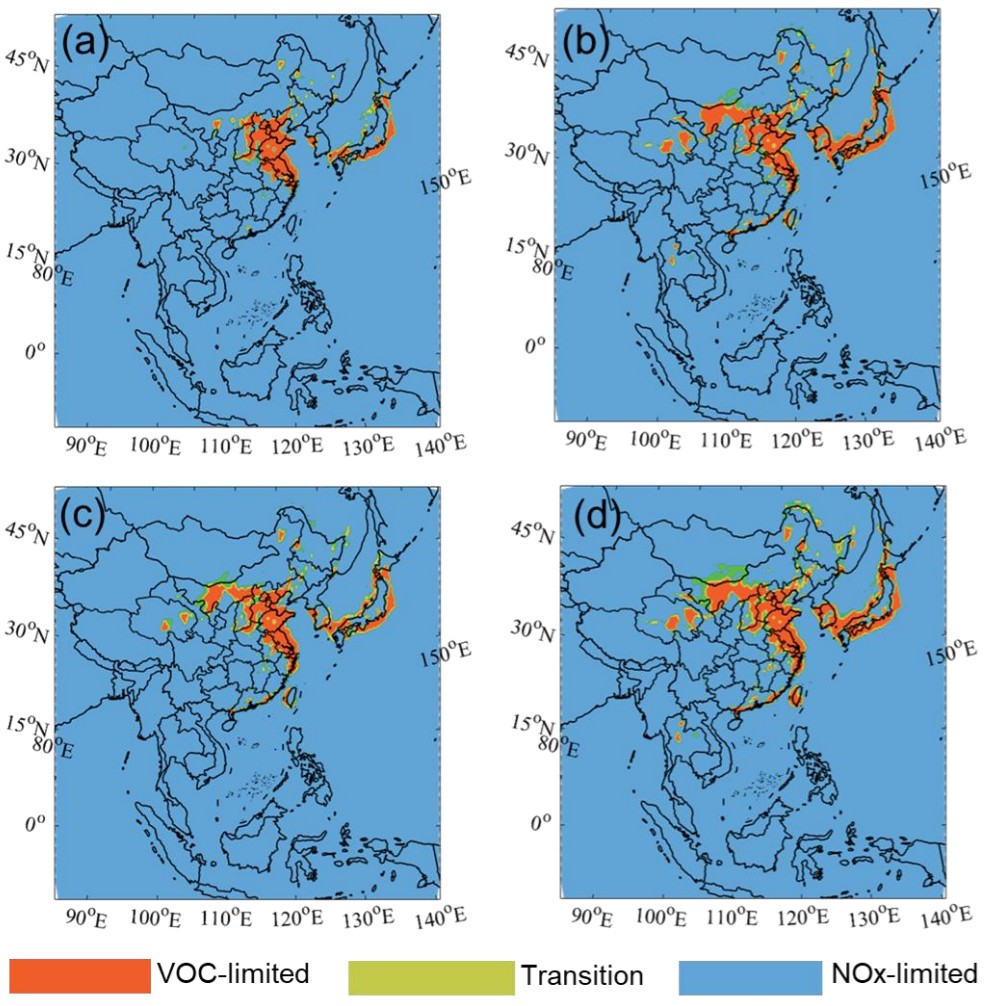

**Figure 7: O₃ sensitivity regimes using (a) Def, (b) HONO, (c) Cl, and (d) Base cases. Classification of ozone sensitivity regime is based on production rates of H₂O₂ to HNO₃, and P$_{H2O2}$/P$_{HNO3}$ of <0.06, 0.06 to 0.2, and >0.2 correspond to VOC-limited, transition, and NOₓ-limited conditions, respectively (Zhang et al., 2009).**


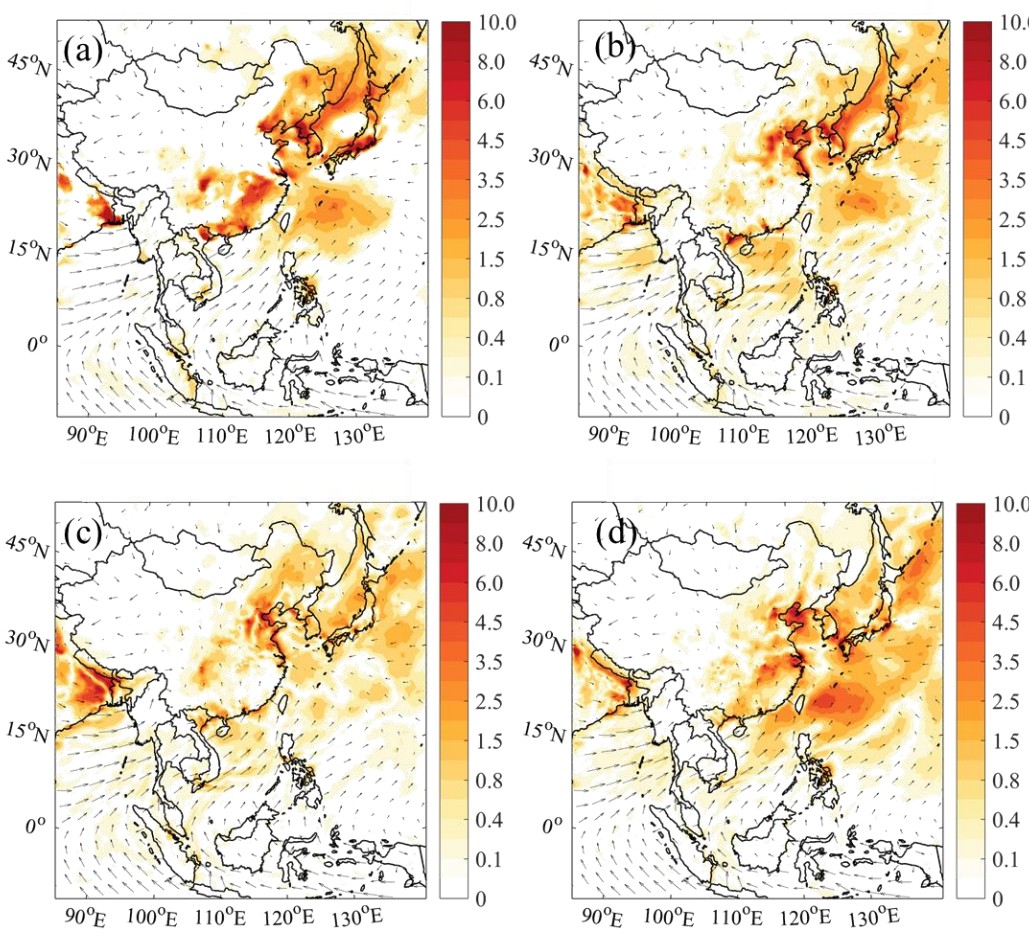

**Figure 8: This figure is the same as Figure 4 but for average PM$_{2.5}$ enhancements (Unit: µg m$^{-3}$).**



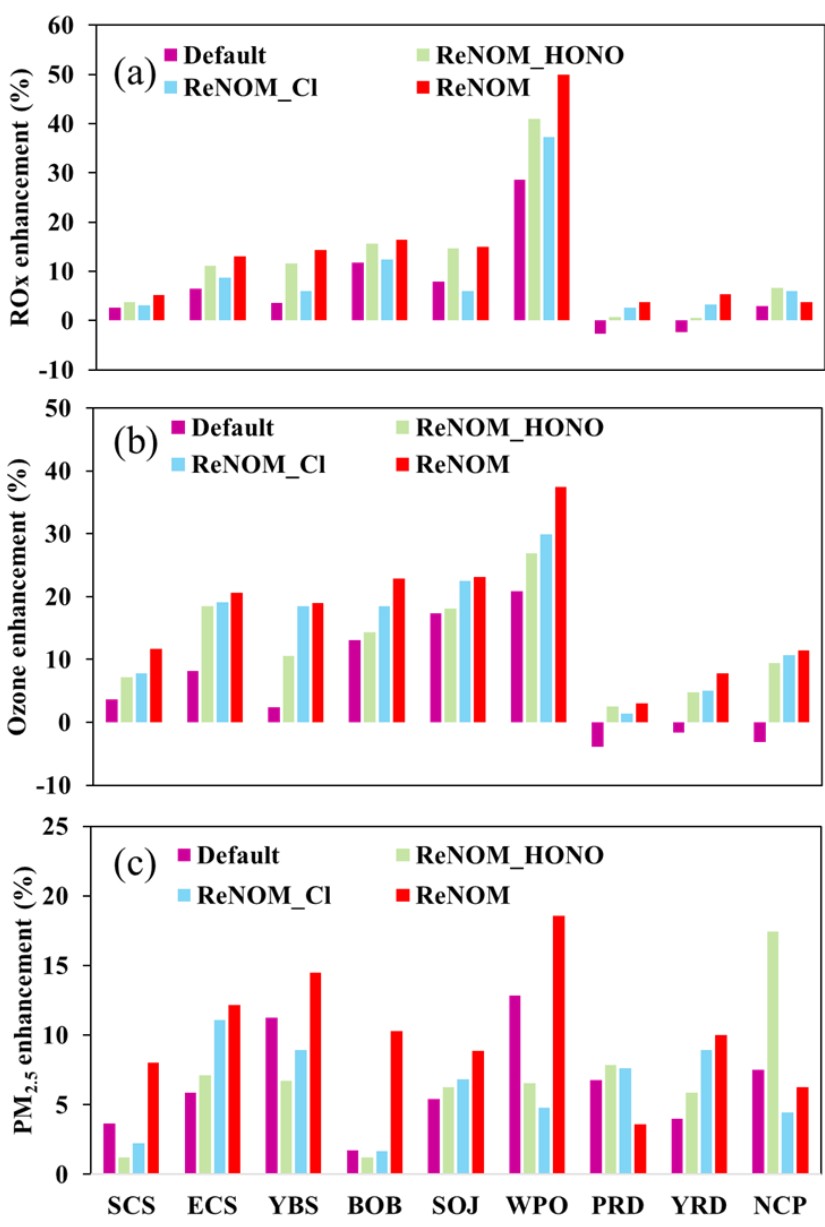


**Figure 9: Contributions of ship emissions with different chemistry to average mixing ratios of (a) daytime RO$_x$, (b) ozone, and (c) PM$_{2.5}$ (with ship case (i.e., Def) – no ship case (i.e., Def_noship)) / with ship case (i.e., Def)).**






**Table 1: Experimental Setting**

| Cases | Anth Emis[a] | Ship Emis[b] | HONO Chem[c] | Chlorine Chem[d] |
|---|---|---|---|---|
| Def | Yes | Yes | No | No |
| Def_noship | Yes | No | No | No |
| Cl | Yes | Yes | No | Yes |
| Cl_noship | Yes | No | No | Yes |
| HONO | Yes | Yes | Yes | No |
| HONO_noship | Yes | No | Yes | No |
| BASE | Yes | Yes | Yes | Yes |
| BASE_noship | Yes | No | Yes | Yes |


[a] Anthropogenic emissions except for ship emissions.

[b] Ship emissions except for directly emitted HONO.

[c] HONO chemistry.

[g] Chlorine chemistry.
