# Peer review of "Impact of International Shipping Emissions on Ozone and PM2.5 in East Asia during Summer: The Important Role of HONO and ClNO2"

_Atmospheric Chemistry and Physics, 2020_

## Author Comment (AC1)

**Response to comments by referees**

We thank the referee for the helpful comments and suggestions. Below are the detailed responses. The referee's comments are in *italic*; our responses are in red.

Reviewer #1

*This paper presents a modeling study on the impact of HONO and ClNO$_2$ chemistry on RO$_x$ budgets and pollutant formation in marine and coastal environments. The WRF-Chem model, with an updated chemical mechanism, is used to determine how ship emissions and added HONO and Cl chemistry affect RO$_x$, O$_3$, and PM$_{2.5}$ levels. The results of the study are clear, and the paper is publishable, however the comments below should be addressed before publication.*

*1. In the model setting section (2.1), more thorough descriptions of the model updates are needed to assess this study. A list of the HONO reactions and their reaction rates would be helpful to readers. The rate should especially be included for the HONO formation from particle nitrate photolysis since this is not included in Zhang et al., 2017, so it is unclear what values are used here.*

Response: Thank you for your suggestion. We updated the model with the additional source of HONO from the photolysis of particular nitrate. The reaction and the photolysis rate of particular nitrate in were added section 2.1, and is also shown below:

"For this study, an additional HONO source from the photolysis of particulate nitrate (PNO$_3$→ 0.67HONO + 0.33 NO$_2$) was updated into our model. The photolysis rate constant of PNO$_3$ ($J_{PNO3}$) followed the reported value used in Fu et al., (2019) ($J_{PNO3}$ = (8.3×10$^{-5}$/7×10$^{-7}$) × $J_{HNO3-WRF-Chem}$; $J_{HNO3-WRF-Chem}$ is the photolysis rate of gaseous HNO$_3$ calculated online in the WRF-Chem model)".

*2. In the emissions section, please cite the land-based HONO/NO$_x$ emission ratios used. It would be useful to give an approximate range of ship-based NO$_x$ emission rates as well since this plays a large role in HONO and ClNO$_2$ chemistry.*

Response: Thank you for your suggestion. We added two citations (Kurtenbach et al., 2001, Gutzwiller et al., 2002) to describe the land-based HONO/NO$_x$ emission ratios used in this study. For the ship-based NO$_x$ emission rates, we showed a spatial distribution of the NO$_x$ emission fluxes from shipping emission inventories (see Figure 1a in the manuscript). Also shown below:

[Figure]

**Figure 1: NOₓ emission fluxes from ships (Unit: g m⁻² month⁻¹) in July 2017.**

*3. In the model validation section (2.4), please clarify what data is being used to validate model performance and which model run is being compared. Are the values listed in Table S2 daytime hourly averages and is this averaged over the entire observational region? Is the SIM values listed for the BASE model case? Similar clarification is needed for Table S3 and the subscript description is incorrect in this table.*

Response: We used the meteorological data, including wind speed, relative humidity, and temperature, from NOAA's National Climatic Data Center and the air pollutants data, including $NO_2$, $PM_{2.5}$, and $O_3$, from China's Ministry of Ecology and Environment to validate the model performance. The simulated data from the BASE case is compared. The values listed in Table 2S are the hourly averages of the available observational data over entire China for the model validation. SIM represents the simulated data from the BASE case. We have revised the table caption in Table S2 and Table S3 as below:

"OBS represents the hourly averages for the available observational data (over the entire regions for the meteorological parameters and over mainland China for the air pollutants). SIM represents the simulations from the BASE case."

*4. In the results section, clarification is again needed about the data that is shown in the figures. Is average HONO referring to daytime averages or 24-hour daily? The values seem quite high if nighttime data is included in the averages.*

Response: the average HONO in the manuscript is the 24-hour daily average. We have clarified this. With the consideration of heterogeneous conversion from $NO_x$ to HONO, the nighttime value of HONO is higher than the value during daytime. For this reason, the relatively high value of HONO in our study is reasonable.

*5. Check the order of your subsections – section 3.2 is missing.*

Response: Thanks for pointing out this problem. It has been corrected.

*6. In line 279, you discuss the switch between $NO_x$ and VOC-sensitive regimes, stating that increased HONO provides an additional source of $NO_x$. The increase in $RO_x$ will also increase the back reaction from NO to HONO. Can you comment on the balance between these two reactions?*

Response: I think the photolysis of HONO is the main reaction in this balance. The back reaction from NO and $RO_x$ to HONO relies on the level of NO and oxidants, and the reaction rate of the photolysis of HONO is faster than that of the back reaction from NO to HONO.

*7. In section 2.4, you state that the model under predicts $NO_2$ and over predicts $PM_{2.5}$. I think this should be discussed in the results/discussion as well as to how this impacts your conclusions about the importance of HONO and Cl chemistry.*

Response: Thank you for your comment. We added a statement in discussion to point out this impact on the importance of HONO and Cl chemistry. And also showed below:

"In our simulation, the under predicted $NO_2$ may lead to the underestimation in HONO, and the over predicted $PM_{2.5}$ can also result in the underestimated uptake of $N_2O5$ and overestimated conversion to HONO. These results may contribute to a further increase in the contribution to average ozone formation and a decrease to average $PM_{2.5}$."

*8. In line 196, you state that HONO spatial distribution is consistent with $NO_2$ due to the homogeneous and heterogeneous conversion. Are you referring to the $HO_2+NO_2$ as the homogeneous conversion? It's my understanding that this is a relatively unimportant HONO source compared to others. A comparison of the default to base run should provide more information since $HO_2+NO_2$ is included in the default mechanism. Perhaps you should discuss if direct emissions of HONO from ships is relevant here.*

Response: Thank you for your comment. We agree that the homogeneous conversion of $NO_x$ and $HO_x$ is a relatively unimportant HONO source and direct emission from ships is relevant. We have changed the statement to "The distribution of HONO was consistent with that of $NO_2$ due to the heterogeneous conversion of $NO_2$ to form HONO and direct HONO emission by ships".

*9. In line 340, the conclusions would be clearer if you presented values for coastal versus oceanic regions rather than just giving the total range of $RO_x$, $O_3$, and $PM_{2.5}$ increases.*

Response: Thanks for the suggestion. We have added the value for coastal and oceanic regions and changed the statement to "The results show that photolysis of the two compounds releases OH and Cl radicals, recycles $NO_x$, and increases conventional hydroxyl and organic peroxy radicals ($RO_x$ = OH + $HO_2$ + $RO_2$) by 0.8% to 21.4% (0.8-7.7% over coast and 2.6-21.4% over ocean), $O_3$ by 5.9% to 16.6% (6.9-14.6% over coast and 5.9-16.6% over ocean), and $PM_{2.5}$ by -1.2% to 8.6% (-1.2-6% over coast and 3.2-8.6% over ocean) at the surface in coastal and Western Pacific regions.".

---

## Author Comment (AC2)

**Response to comments by referees**

We thank the referee for the helpful comments and suggestions. Below are the detailed responses. The referee's comments are in *italic*; our responses are in red.

Reviewer #2

*The effects of ship emissions on the formation of $O_3$ and $PM_{2.5}$ have a significant impact on the climate, air quality, and human health. However, limited attention has been paid to the production of ship-related radicals in evaluating the effects of ship emissions on secondary pollutants. This study used a revised regional chemical transport model (CBMZ was updated to CBMZ-ReNOM) to simulate the spatial distributions of HONO and $ClNO_2$ produced by ocean-going ships and their effects on the formation of $O_3$ and $PM_{2.5}$. Overall, this is a fundamental work with clear importance. It fulfils the necessary requirements to be published. I recommend it for publication after the authors consider several minor revisions to the manuscript.*

*1. The model simulations were performed from June 28 to July 31, 2018. It's the summertime for east Asia. Can you expect what's the change of main conclusions if you expand the simulation to all seasons? If it's hard to expect the results for different seasons, the title should be specified to summer.*

Response: Thank you for your comment. We chose summer for this study because the relative impact of ship emission may be most distinctive in the western Pacific due to smallest influence of land emissions under large-scale winds from oceans. Moreover, high temperature and strong solar radiations during summer lead to the fast production of ozone and other secondary pollutants. However, it is difficult to expect the quantitative impact for other seasons. We have revised the title to indicate our work is for summer:

"Impact of International Shipping Emissions on Ozone and $PM_{2.5}$ in East Asia during Summer: The Important Role of HONO and $ClNO_2$".

*2. The HONO emissions from land transportation sources were calculated using land-based NOx emissions and the HONO/NOx ratios (0.8% for gasoline and 2.3% for diesel). It should be noted that the estimation is quite rough. It would be useful to give a range of HONO and check the impacts.*

Response: The emission ratio of 0.8% for gasoline and 2.3% for diesel are based on the previous experiments studies (Kurtenbach et al., 2001, Gutzwiller et al., 2002) and have been widely used in model studies(Zhang et al., 2016, Sun et al., 2020, Fu et al., 2019). These ratios are generally consistent with more recent measurements (Liu et al., 2019, Trinh et al., 2017). We believe these

ratios are reasonable. We have cited the two papers (Kurtenbach et al., 2001, Gutzwiller et al., 2002) in the emissions section to explain the reasons for using these HONO/NO$_x$ ratios in this study.

*3. The underpredicted O$_3$ on land is larger than on maritime regions. Are there any correlations between the two? If so, is the ReNOM scheme still important?*

Response: the larger underpredicted O$_3$ in land area was related to the higher absolute value of O$_3$ in this area. Considering the different sources of ozone precursors in land and marine area, it is difficult to correlate the underpredicted ozone in these two regions. In our model results, the simulated ozone was improved by ReNOM scheme (with smaller bias) in both land and marine sites. We believe that the consideration of ReNOM scheme is important to improve the ozone simulation in these two areas.

*4. Fig. 2. Both of the concentrations of HONO and ClNO$_2$ are very low on the ocean. How can you determine the contribution from ships is accurate, not noise from the model?*

Response: Thank you for your comment. According to previous studies, the observed levels of HONO and ClNO$_2$ in remote oceans are low, 3-35 pptv and 89 pptv, respectively (Ye et al., 2016, Kasibhatla et al., 2018, Meusel et al., 2016). Our simulated results for these two species are consistent with the observed values. Moreover, the simulated HONO and ClNO$_2$ (especially HONO) over marine areas were consistent with the distribution of ship tracks. We also repeated the simulations for the same model runs and obtained the consistent results on HONO and ClNO$_2$. Therefore we believe that our simulated HONO and ClNO$_2$ in marine regions is mainly from ship emissions, not the noise from the model.

*5. Fig.6d and 8d show a hot spot in inland area of south China. As the inland river ship emissions were not included in this study, how to explain the reason for the most significant changes happened in inland, which is isolated from shipping emissions? In another words, if other reasons would drive to such high increment, how to confirm the other increments are from ships not noise?*

Response: Thank you for your comment. We also noticed some ship-induced hot spots in the inland areas. To check the accuracy of our model results, we had re-run the BASE and Default case with the same model setting. The hot spots in the inland areas of south China remained. The hot spots may be a result of inhomogeneous impact of ship emissions due to complicated dynamic and chemical processes that affect the fate and distribution of ship-emitted pollutants in the inland areas. In particular, the mountainous terrains in south China may have large influence on transport of ship emissions to the inland areas. We have added the below discussion in the revised version.

"In addition to the above coastal and oceanic areas, ship emissions also exert considerable impact on surface O3 in distant inland areas such as Sichuan basin, and interestingly there are some 'hot spots' of ozone increase/decrease in the inland areas due to ship emissions (Figure 6a-d) (as well as ROx (Figure 4a-d) and PM2.5 (Figure 8a-d)). These hot spots may be a result of inhomogeneous impact of ship emissions due to complicated dynamic and chemical processes that affect the fate and distribution of ship-emitted pollutants in the inland areas. In particular, the mountainous terrains in south China may have large influence on transport of ship emissions to the inland areas."

*6. Current titles for Fig. 6 and 8 are not appropriate.*

Response: We changed the title for Fig. 6 to "24-hour daily averaged ozone variations (06:00-18:00 LST; Unit: ppbv) with (a) default chemistry (Def-Def_noship), (b) default and additional HONO chemistry (HONO-HONO_noship), (c) default and additional chlorine chemistry (Cl-Cl_noship), and (d) default and combined HONO and chlorine chemistry (BASE-BASE_noship). Arrows present simulated wind vectors from BASE case.". The current title for Fig.8 changed to "Averaged PM$_{2.5}$ enhancements (Unit: µg m$^{-3}$) with (a) default chemistry (Def-Def_noship), (b) default and additional HONO chemistry (HONO-HONO_noship), (c) default and additional chlorine chemistry (Cl-Cl_noship), and (d) default and combined HONO and chlorine chemistry (BASE-BASE_noship). Arrows present simulated wind vectors from BASE case"

*7. Section 3.2 and title for section 3.3 are missing.*

Response: Thanks for pointing out this effort. It has been corrected.

References:

FU, X., WANG, T., ZHANG, L., LI, Q., WANG, Z., XIA, M., YUN, H., WANG, W., YU, C. & YUE, D. 2019. The significant contribution of HONO to secondary pollutants during a severe winter pollution event in southern China.

GUTZWILLER, L., ARENS, F., BALTENSPERGER, U., GÄGGELER, H. W. & AMMANN, M. 2002. Significance of semivolatile diesel exhaust organics for secondary HONO formation. *Environmental science & technology,* 36**,** 677-682.

KASIBHATLA, P., SHERWEN, T., EVANS, M. J., CARPENTER, L. J., REED, C., ALEXANDER, B., CHEN, Q., SULPRIZIO, M. P., LEE, J. D. & READ, K. A. 2018. Global impact of nitrate photolysis in sea-salt

aerosol on NOx, OH, and O3 in the marine boundary layer. *Atmospheric Chemistry and Physics*, 11185-11203.

KURTENBACH, R., BECKER, K., GOMES, J., KLEFFMANN, J., LÖRZER, J., SPITTLER, M., WIESEN, P., ACKERMANN, R., GEYER, A. & PLATT, U. 2001. Investigations of emissions and heterogeneous formation of HONO in a road traffic tunnel. *Atmospheric Environment,* 35, 3385-3394.

LIU, Y., NIE, W., XU, Z., WANG, T., WANG, R., LI, Y., WANG, L., CHI, X. & DING, A. 2019. Contributions of different sources to nitrous acid (HONO) at the SORPES station in eastern China: results from one-year continuous observation. *Atmos. Chem. Phys. Discuss*, 1-47.

MEUSEL, H., KUHN, U., REIFFS, A., MALLIK, C., HARDER, H., MARTINEZ, M., SCHULADEN, J., BOHN, B., PARCHATKA, U. & CROWLEY, J. N. 2016. Daytime formation of nitrous acid at a coastal remote site in Cyprus indicating a common ground source of atmospheric HONO and NO. *Atmospheric Chemistry and Physics,* 16, 14475-14493.

SUN, L., CHEN, T., JIANG, Y., ZHOU, Y., SHENG, L., LIN, J., LI, J., DONG, C., WANG, C. & WANG, X. 2020. Ship emission of nitrous acid (HONO) and its impacts on the marine atmospheric oxidation chemistry. *Science of The Total Environment*, 139355.

TRINH, H. T., IMANISHI, K., MORIKAWA, T., HAGINO, H. & TAKENAKA, N. 2017. Gaseous nitrous acid (HONO) and nitrogen oxides (NOx) emission from gasoline and diesel vehicles under real-world driving test cycles. *Journal of the Air & Waste Management Association,* 67, 412-420.

YE, C., ZHOU, X., PU, D., STUTZ, J., FESTA, J., SPOLAOR, M., TSAI, C., CANTRELL, C., MAULDIN, R. L. & CAMPOS, T. 2016. Rapid cycling of reactive nitrogen in the marine boundary layer. *Nature,* 532, 489-491.

ZHANG, L., WANG, T., ZHANG, Q., ZHENG, J., XU, Z. & LV, M. 2016. Potential sources of nitrous acid (HONO) and their impacts on ozone: A WRF‐Chem study in a polluted subtropical region. *Journal of Geophysical Research: Atmospheres,* 121, 3645-3662.

---

## Author Comment (AC3)

**Response to comments by referees**

We thank the referee for the helpful comments and suggestions. Below are the detailed responses. The referee's comments are in *italic*; our responses are in red.

Reviewer #1

*This paper presents a modeling study on the impact of HONO and ClNO$_2$ chemistry on RO$_x$ budgets and pollutant formation in marine and coastal environments. The WRF-Chem model, with an updated chemical mechanism, is used to determine how ship emissions and added HONO and Cl chemistry affect RO$_x$, O$_3$, and PM$_{2.5}$ levels. The results of the study are clear, and the paper is publishable, however the comments below should be addressed before publication.*

*1. In the model setting section (2.1), more thorough descriptions of the model updates are needed to assess this study. A list of the HONO reactions and their reaction rates would be helpful to readers. The rate should especially be included for the HONO formation from particle nitrate photolysis since this is not included in Zhang et al., 2017, so it is unclear what values are used here.*

Response: Thank you for your suggestion. We updated the model with the additional source of HONO from the photolysis of particular nitrate. The reaction and the photolysis rate of particular nitrate in were added section 2.1, and is also shown below:

"For this study, an additional HONO source from the photolysis of particulate nitrate (PNO$_3 \rightarrow$ 0.67HONO + 0.33 NO$_2$) was updated into our model. The photolysis rate constant of PNO$_3$ (J$_{PNO3}$) followed the reported value used in Fu et al., (2019) (J$_{PNO3}$ = (8.3×10$^{-5}$/7×10$^{-7}$) × J$_{HNO3-WRF-Chem}$; J$_{HNO3-WRF-Chem}$ is the photolysis rate of gaseous HNO$_3$ calculated online in the WRF-Chem model)".

*2. In the emissions section, please cite the land-based HONO/NO$_x$ emission ratios used. It would be useful to give an approximate range of ship-based NO$_x$ emission rates as well since this plays a large role in HONO and ClNO$_2$ chemistry.*

Response: Thank you for your suggestion. We added two citations (Kurtenbach et al., 2001, Gutzwiller et al., 2002) to describe the land-based HONO/NO$_x$ emission ratios used in this study. For the ship-based NO$_x$ emission rates, we showed a spatial distribution of the NO$_x$ emission fluxes from shipping emission inventories (see Figure 1a in the manuscript). Also shown below:

[Figure]

**Figure 1: NOₓ emission fluxes from ships (Unit: g m⁻² month⁻¹) in July 2017.**

*3. In the model validation section (2.4), please clarify what data is being used to validate model performance and which model run is being compared. Are the values listed in Table S2 daytime hourly averages and is this averaged over the entire observational region? Is the SIM values listed for the BASE model case? Similar clarification is needed for Table S3 and the subscript description is incorrect in this table.*

Response: We used the meteorological data, including wind speed, relative humidity, and temperature, from NOAA's National Climatic Data Center and the air pollutants data, including $NO_2$, $PM_{2.5}$, and $O_3$, from China's Ministry of Ecology and Environment to validate the model performance. The simulated data from the BASE case is compared. The values listed in Table 2S are the hourly averages of the available observational data over entire China for the model validation. SIM represents the simulated data from the BASE case. We have revised the table caption in Table S2 and Table S3 as below:

"OBS represents the hourly averages for the available observational data (over the entire regions for the meteorological parameters and over mainland China for the air pollutants). SIM represents the simulations from the BASE case."

*4. In the results section, clarification is again needed about the data that is shown in the figures. Is average HONO referring to daytime averages or 24-hour daily? The values seem quite high if nighttime data is included in the averages.*

Response: the average HONO in the manuscript is the 24-hour daily average. We have clarified this. With the consideration of heterogeneous conversion from $NO_x$ to HONO, the nighttime value of HONO is higher than the value during daytime. For this reason, the relatively high value of HONO in our study is reasonable.

*5. Check the order of your subsections – section 3.2 is missing.*

Response: Thanks for pointing out this problem. It has been corrected.

*6. In line 279, you discuss the switch between $NO_x$ and VOC-sensitive regimes, stating that increased HONO provides an additional source of $NO_x$. The increase in $RO_x$ will also increase the back reaction from NO to HONO. Can you comment on the balance between these two reactions?*

Response: I think the photolysis of HONO is the main reaction in this balance. The back reaction from NO and $RO_x$ to HONO relies on the level of NO and oxidants, and the reaction rate of the photolysis of HONO is faster than that of the back reaction from NO to HONO.

*7. In section 2.4, you state that the model under predicts $NO_2$ and over predicts $PM_{2.5}$. I think this should be discussed in the results/discussion as well as to how this impacts your conclusions about the importance of HONO and Cl chemistry.*

Response: Thank you for your comment. We added a statement in discussion to point out this impact on the importance of HONO and Cl chemistry. And also showed below:

"In our simulation, the under predicted $NO_2$ may lead to the underestimation in HONO, and the over predicted $PM_{2.5}$ can also result in the underestimated uptake of $N_2O5$ and overestimated conversion to HONO. These results may contribute to a further increase in the contribution to average ozone formation and a decrease to average $PM_{2.5}$."

*8. In line 196, you state that HONO spatial distribution is consistent with $NO_2$ due to the homogeneous and heterogeneous conversion. Are you referring to the $HO_2+NO_2$ as the homogeneous conversion? It's my understanding that this is a relatively unimportant HONO source compared to others. A comparison of the default to base run should provide more information since $HO_2+NO_2$ is included in the default mechanism. Perhaps you should discuss if direct emissions of HONO from ships is relevant here.*

Response:  Thank you for your comment. We agree that the homogeneous conversion of $NO_x$ and $HO_x$ is a relatively unimportant HONO source and direct emission from ships is relevant. We have changed the statement to "The distribution of HONO was consistent with that of $NO_2$ due to the heterogeneous conversion of $NO_2$ to form HONO and direct HONO emission by ships".

9. *In line 340, the conclusions would be clearer if you presented values for coastal versus oceanic regions rather than just giving the total range of $RO_x$, $O_3$, and $PM_{2.5}$ increases.*

Response: Thanks for the suggestion. We have changed the statement to "The results show that photolysis of the two compounds releases OH and Cl radicals, recycles $NO_x$, and increases conventional hydroxyl and organic peroxy radicals ($RO_x$ = OH + $HO_2$ + $RO_2$) by 0.8% to 21.4% (0.8-7.7% over coast and 2.6-21.4% over ocean), $O_3$ by 5.9% to 16.6% (6.9-14.6% over coast and 5.9-16.6% over ocean), and $PM_{2.5}$ by -1.2% to 8.6% (-1.2-6% over coast and 3.2-8.6% over ocean) at the surface in coastal and Western Pacific regions.".

Reviewer #2

*The effects of ship emissions on the formation of $O_3$ and $PM_{2.5}$ have a significant impact on the climate, air quality, and human health. However, limited attention has been paid to the production of ship-related radicals in evaluating the effects of ship emissions on secondary pollutants. This study used a revised regional chemical transport model (CBMZ was updated to CBMZ-ReNOM) to simulate the spatial distributions of HONO and $ClNO_2$ produced by ocean-going ships and their effects on the formation of $O_3$ and $PM_{2.5}$. Overall, this is a fundamental work with clear importance. It fulfils the necessary requirements to be published. I recommend it for publication after the authors consider several minor revisions to the manuscript.*

1. *The model simulations were performed from June 28 to July 31, 2018. It's the summertime for east Asia. Can you expect what's the change of main conclusions if you expand the simulation to all seasons? If it's hard to expect the results for different seasons, the title should be specified to summer.*

Response: Thank you for your comment. We chose summer for this study because the relative impact of ship emission may be most distinctive in the western Pacific due to smallest influence of land emissions under large-scale winds from oceans.  Moreover, high temperature and strong solar radiations during summer lead to the fast production of ozone and other secondary pollutants. However, it is difficult to expect the quantitative impact for other seasons. We have revised the title to indicate our work is for summer:

"Impact of International Shipping Emissions on Ozone and PM$_{2.5}$ in East Asia during Summer: The Important Role of HONO and ClNO$_2$".

*2. The HONO emissions from land transportation sources were calculated using land-based NOx emissions and the HONO/NOx ratios (0.8% for gasoline and 2.3% for diesel). It should be noted that the estimation is quite rough. It would be useful to give a range of HONO and check the impacts.*

Response: The emission ratio of 0.8% for gasoline and 2.3% for diesel are based on the previous experiments studies (Kurtenbach et al., 2001, Gutzwiller et al., 2002) and have been widely used in model studies(e.g., Zhang et al., 2016, Sun et al., 2020, Fu et al., 2019). These ratios are generally consistent with more recent measurements (Liu et al., 2019, Trinh et al., 2017). We believe these ratios are reasonable. We have cited the two papers (Kurtenbach et al., 2001, Gutzwiller et al., 2002) in the emissions section to explain the reasons for using these HONO/NO$_x$ ratios in this study.

*3. The underpredicted O$_3$ on land is larger than on maritime regions. Are there any correlations between the two? If so, is the ReNOM scheme still important?*

Response: the larger underpredicted O$_3$ in land area was related to the higher absolute value of O$_3$ in this area. Considering the different sources of ozone precursors in land and marine area, it is difficult to correlate the underpredicted ozone in these two regions. In our model results, the simulated ozone was improved by ReNOM scheme (with smaller bias) in both land and marine sites. We believe that the consideration of ReNOM scheme is important to improve the ozone simulation in these two areas.

*4. Fig. 2. Both of the concentrations of HONO and ClNO$_2$ are very low on the ocean. How can you determine the contribution from ships is accurate, not noise from the model?*

Response: Thank you for your comment. According to previous studies, the observed levels of HONO and ClNO$_2$ in remote oceans are low, 3-35 pptv and 89 pptv, respectively (Ye et al., 2016, Kasibhatla et al., 2018, Meusel et al., 2016). Our simulated results for these two species are consistent with the observed values. Moreover, the simulated HONO and ClNO$_2$ (especially HONO) over marine areas were consistent with the distribution of ship tracks. We also repeated the simulations for the same model runs and obtained the consistent results on HONO and ClNO$_2$. Therefore we believe that our simulated HONO and ClNO$_2$ in marine regions is mainly from ship emissions, not the noise from the model.

*5. Fig.6d and 8d show a hot spot in inland area of south China. As the inland river ship emissions were not included in this study, how to explain the reason for the most significant changes happened in inland, which is isolated from shipping emissions? In another words, if other reasons would drive to such high increment, how to confirm the other increments are from ships not noise?*

Response:  Thank you for your comment. We also noticed some ship-induced hot spots in the inland areas. To check the accuracy of our model results, we had re-run the BASE and Default case with the same model setting. The hot spots in the inland areas of south China remained. The hot spots may be a result of inhomogeneous impact of ship emissions due to complicated dynamic and chemical processes that affect the fate and distribution of ship-emitted pollutants in the inland areas.  In particular, the mountainous terrains in south China may have large influence on transport of ship emissions to the inland areas.  We have added the below discussion in the revised version.

"In addition to the above coastal and oceanic areas, ship emissions also exert considerable impact on surface O3 in distant inland areas such as Sichuan basin, and interestingly there are some 'hot spots' of ozone increase/decrease in the inland areas due to ship emissions (Figure 6a-d) (as well as ROx (Figure 4a-d) and PM2.5 (Figure 8a-d)). These hot spots may be a result of inhomogeneous impact of ship emissions due to complicated dynamic and chemical processes that affect the fate and distribution of ship-emitted pollutants in the inland areas.  In particular, the mountainous terrains in south China may have large influence on transport of ship emissions to the inland areas."

*6. Current titles for Fig. 6 and 8 are not appropriate.*

Response: We changed the title for Fig. 6 to "24-hour daily averaged ozone variations (06:00-18:00 LST; Unit: ppbv) with (a) default chemistry (Def-Def_noship), (b) default and additional HONO chemistry (HONO-HONO_noship), (c) default and additional chlorine chemistry (Cl-Cl_noship), and (d) default and combined HONO and chlorine chemistry (BASE-BASE_noship). Arrows present simulated wind vectors from BASE case.". The current title for Fig.8 changed to "Averaged PM2.5 enhancements (Unit: $\mu g\ m^{-3}$) with (a) default chemistry (Def-Def_noship), (b) default and additional HONO chemistry (HONO-HONO_noship), (c) default and additional chlorine chemistry (Cl-Cl_noship), and (d) default and combined HONO and chlorine chemistry (BASE-BASE_noship). Arrows present simulated wind vectors from BASE case"

*7. Section 3.2 and title for section 3.3 are missing.*

Response: Thanks for pointing out this effort. It has been corrected.

References:

FU, X., WANG, T., ZHANG, L., LI, Q., WANG, Z., XIA, M., YUN, H., WANG, W., YU, C. & YUE, D. 2019. The significant contribution of HONO to secondary pollutants during a severe winter pollution event in southern China.

GUTZWILLER, L., ARENS, F., BALTENSPERGER, U., GÄGGELER, H. W. & AMMANN, M. 2002. Significance of semivolatile diesel exhaust organics for secondary HONO formation. *Environmental science & technology,* 36**,** 677-682.

KASIBHATLA, P., SHERWEN, T., EVANS, M. J., CARPENTER, L. J., REED, C., ALEXANDER, B., CHEN, Q., SULPRIZIO, M. P., LEE, J. D. & READ, K. A. 2018. Global impact of nitrate photolysis in sea-salt aerosol on NOx, OH, and O3 in the marine boundary layer. *Atmospheric Chemistry and Physics***,** 11185-11203.

KURTENBACH, R., BECKER, K., GOMES, J., KLEFFMANN, J., LÖRZER, J., SPITTLER, M., WIESEN, P., ACKERMANN, R., GEYER, A. & PLATT, U. 2001. Investigations of emissions and heterogeneous formation of HONO in a road traffic tunnel. *Atmospheric Environment,* 35**,** 3385-3394.

LIU, H., JIN, X., WU, L., WANG, X., FU, M., LV, Z., MORAWSKA, L., HUANG, F. & HE, K. 2018. The impact of marine shipping and its DECA control on air quality in the Pearl River Delta, China. *Science of The Total Environment,* 625**,** 1476-1485.

LIU, Y., NIE, W., XU, Z., WANG, T., WANG, R., LI, Y., WANG, L., CHI, X. & DING, A. 2019. Contributions of different sources to nitrous acid (HONO) at the SORPES station in eastern China: results from one-year continuous observation. *Atmos. Chem. Phys. Discuss***,** 1-47.

LV, Z., LIU, H., YING, Q., FU, M., MENG, Z., WANG, Y., WEI, W., GONG, H. & HE, K. 2018. Impacts of shipping emissions on PM2: 5 pollution in China. *Atmospheric Chemistry & Physics,* 18.

MEUSEL, H., KUHN, U., REIFFS, A., MALLIK, C., HARDER, H., MARTINEZ, M., SCHULADEN, J., BOHN, B., PARCHATKA, U. & CROWLEY, J. N. 2016. Daytime formation of nitrous acid at a coastal remote site in Cyprus indicating a common ground source of atmospheric HONO and NO. *Atmospheric Chemistry and Physics,* 16**,** 14475-14493.

SUN, L., CHEN, T., JIANG, Y., ZHOU, Y., SHENG, L., LIN, J., LI, J., DONG, C., WANG, C. & WANG, X. 2020. Ship emission of nitrous acid (HONO) and its impacts on the marine atmospheric oxidation chemistry. *Science of The Total Environment***,** 139355.

TRINH, H. T., IMANISHI, K., MORIKAWA, T., HAGINO, H. & TAKENAKA, N. 2017. Gaseous nitrous acid (HONO) and nitrogen oxides (NOx) emission from gasoline and diesel vehicles under real-world driving test cycles. *Journal of the Air & Waste Management Association,* 67**,** 412-420.

WANG, P., QIAO, X. & ZHANG, H. 2020. Modeling PM2. 5 and O3 with aerosol feedbacks using WRF/Chem over the Sichuan Basin, southwestern China. *Chemosphere,* 254**,** 126735.

YE, C., ZHOU, X., PU, D., STUTZ, J., FESTA, J., SPOLAOR, M., TSAI, C., CANTRELL, C., MAULDIN, R. L. & CAMPOS, T. 2016. Rapid cycling of reactive nitrogen in the marine boundary layer. *Nature,* 532**,** 489-491.

ZHANG, L., WANG, T., ZHANG, Q., ZHENG, J., XU, Z. & LV, M. 2016. Potential sources of nitrous acid (HONO) and their impacts on ozone: A WRF-Chem study in a polluted subtropical region. *Journal of Geophysical Research: Atmospheres,* 121**,** 3645-3662.